# Forecasting Hospital Readmissions with Machine Learning

**DOI:** 10.3390/healthcare10060981

**Published:** 2022-05-25

**Authors:** Panagiotis Michailidis, Athanasia Dimitriadou, Theophilos Papadimitriou, Periklis Gogas

**Affiliations:** 1Department of Economics, Democritus University of Thrace, 69100 Komotini, Greece; michailidis.p@outlook.com (P.M.); pgkogkas@econ.duth.gr (P.G.); 2Department of Economics, University of Derby, Derby DE22 1GB, UK; nancy.dimitriadou@gmail.com

**Keywords:** machine learning, forecasting, readmissions

## Abstract

Hospital readmissions are regarded as a compounding economic factor for healthcare systems. In fact, the readmission rate is used in many countries as an indicator of the quality of services provided by a health institution. The ability to forecast patients’ readmissions allows for timely intervention and better post-discharge strategies, preventing future life-threatening events, and reducing medical costs to either the patient or the healthcare system. In this paper, four machine learning models are used to forecast readmissions: support vector machines with a linear kernel, support vector machines with an RBF kernel, balanced random forests, and weighted random forests. The dataset consists of 11,172 actual records of hospitalizations obtained from the General Hospital of Komotini “Sismanogleio” with a total of 24 independent variables. Each record is composed of administrative, medical-clinical, and operational variables. The experimental results indicate that the balanced random forest model outperforms the competition, reaching a sensitivity of 0.70 and an AUC value of 0.78.

## 1. Introduction

In recent decades, researchers have paid special attention to the readmission rate as it is considered a globally attractive and reliable indicator of the effectiveness and quality of hospital care provided to the patients. National health systems in many countries (e.g., the U.S.) began using the readmission rates as a publicly reported metric that can be used for hospital comparison and determination of hospital services reimbursement. Accurate forecasting of hospital readmissions can contribute significantly to the improvement of the quality of services provided by a specific hospital or by a health care system as a whole. The substantial health, economic and social benefits of such forecasting are undeniable. The benefits for the patients are related to the timely diagnosis and effective treatment of a health issue, the prevention of complications due to the delayed or incorrect treatment, and the shortening of hospitalization time while reducing potential psychological consequences. The number of readmissions to the total hospital admissions is a significant ratio, which can be used as a business intelligence (BI) tool to uncover service quality insights and improve patient care. The study of readmission rates may enhance the discharge processes by providing better medication reconciliation and implementing follow-up processes for discharged patients.

Inpatient curative costs constitute the majority of a hospital’s total expenditures. According to the 2019 OECD survey, the average spending on inpatient care services for the 36 OECD countries accounts for approximately 65% of a hospital’s total expenditures [1]. In Greece, a member of the OECD, this figure peaks by reaching 93%. The definition of the term “readmission” ranges in the relevant literature from 1 [2,3,4], 2 [5], 3 [6] to 12 months [7] after first initial admission. Almost 20% of the patients in the U.S. covered by Medicare are readmitted within 30 days of their initial discharge with an estimated annual cost of $17 billion [8]. The readmissions rate as a quality measure of the provided healthcare services has been extensively researched for decades [9,10,11,12]. Identifying the high-risk patients for readmissions, managing medication reconciliation, optimizing the use of technology, and planning a transitional care model, are some of the strategies that can be applied to reduce preventable readmissions.

Obviously, readmission forecasting can prevent some repeatable hospitalizations, though it can also lead to the optimal distribution of available resources and the identification of the readmission risk factors that were previously unidentified. Usually, readmission forecasting is based on traditional statistical forecasting models, such as in the work of Kansagara et al. [13] and Zhou et al. [14].

More recently, there are studies in the relevant literature employing methodological tools from the arsenal of machine learning. Machine learning (ML) is one of the most important and fastest growing branches of artificial intelligence. ML techniques have been successfully applied to various problems, ranging from pattern recognition, computer vision, economics, and finance, to medical applications. The most important property of these methods is their distinctive ability to train an accurate and general model from finite input data.

Li et al. [15] used administrative and demographic data from a data set of 1,631,611 hospitalization records in Quebec during the period 1995 to 2012. The aim of the study, as they mention, was to find the probability of a patient’s readmission within 30 days after the initial discharge. They did so by comparing a traditional logistic regression model with the appropriate penalization and a toolbox of machine learning methods, including random forests, deep learning, and extreme gradient boosting. The findings show that the random forest and extreme gradient boosting algorithms are the most accurate models, achieving an area under the curve (AUC) value of 0.79 at hospital admission and an AUC value above 0.88 at hospital discharge.

Moreover, Wang et al. [12] evaluated the merits of convolutional neural networks (CNN) using the general hospital wards data for 30-day readmission prediction. They found that the area under the curve (AUC) was 0.70, significantly higher than all the baseline methods. Futoma et al. [16] applied a deep neural network (NN) to predict early hospital readmission and achieved better predictive performance in all conditions examined (AUC = 0.73) compared to penalized logistic regression models.

In a recent study, Zhou et al. [17] focused on elderly hospital readmission prediction, proposing a fuzzy partition enhanced weighted factorization machine (WFM) model. The model was tested on a large-scale dataset from public hospitals in Hong Kong, which included elderly (65+) patients with chronic kidney disease (CKD). The proposed WFMFP model outperformed several widely used methods, including standard FM, XGBoost, Lightgbm, gradient boosting machine, random forests, and SVMs with different kernels.

In a meta-research, Mahmoudi et al. [18] collected 41 studies during the period from 2015 to 2019, focusing on the comparison of traditional and ML forecasting models. They found that on average, ML models outperformed statistical ones: the AUC value achieved by the machine learning models was 0.74; the AUC value from traditional regression models was 0.71. However, the difference was not statistically significant. Another recent meta-research by Huang et al. [19] studied 43 ML based studies on forecasting hospital readmissions in the U.S. between 2015 and 2019. According to this, the neural networks and the boosted trees were the most accurate forecasting models.

To the best of our knowledge, there is just one attempt to forecast the readmission rate in Greece [20]. The authors in this paper tested five ML algorithms for the prediction of hospital readmissions using data from the General Hospital “Elpis” in Athens, Greece. The dataset does not use any medical data in the prediction but includes eight administrative variables derived from the hospital information system on 131,872 hospitalization records spanning the period from 2000 to 2017. The authors concluded that the K-nearest neighbors algorithm provided the most accurate model with an AUC = 0.785.

It is worth noting that, based on the recent Greek and international literature studied, only a minority of the research papers reported the application of a method to address imbalanced datasets. Du et al. [21] proposed a new risk prediction method termed joint imbalanced classification and feature selection (JICFS) to overcome the issue of class-imbalance in hospital readmission. Wang et al. [12] proposed a cost-sensitive deep learning model to address the imbalanced problem in medical data. In another study, Du et al. [22] introduced a method combining a graph-based technique, an optimization framework, and a neural network to achieve superior performance in predicting hospital readmissions.

In this paper, we use data from the Sismanogleio Public Hospital in Komotini, Greece. The dataset covers the period from 2018 to 2019. The variables are taken from the hospital information system. All records used were anonymized at the source, and the patients’ identifying information was substituted by a numerical sequence before we received them. We employ four alternative machine learning models: support vector machines with linear kernel, support vector machine with RBF (radial basis function) kernel, weighted random forests, and balanced random forests. In our performed empirical analysis, we considered the problem of the imbalance on our dataset: there is one readmission for every 6.42 cases (1741 readmissions in 11,172 total cases/admissions). We record a readmission whenever a patient is admitted again to the hospital within a period of 30 days after the initial discharge. The initial data set was divided into two sub-samples: the in-sample (90% of our dataset) used to train the models and the out-of-sample (10% of our dataset) used to test their generalization ability. We created the sub-samples using a stratified random sampling technique to preserve the ratio of readmissions observed in the full data set to the two sub-samples. In our setup, we test the performance of our models using both the sensitivity and the AUC metrics.

We must note that this is one of the few studies in the literature that forecasts readmission rates based on three types of real-world data: a) medical-clinical data, b) administrative/demographic data, and c) operational data. To the best of our knowledge, most of the research that is published in this field uses mainly EMR data or a mix of medical data and variables that belong to the domain of demographic factors. We believe that the addition of hospital operational data is an important contribution to this line of literature.

Amongst a total of 84 studies that have been included in the systematic reviews by Mahmoudi et al. [18] and Huang et al. [19], a variety of clinical data, composite clinical scores, disability/functional status measures, demographics, and socioeconomic status features have been used. None of the abovementioned studies have taken advantage of the operational data of the clinics and hospitals and thus have not examined the use of such data to forecast readmissions.

Operational data such as the current clinic’s occupancy rate, clinic’s number of doctors, and clinic’s number of nurses on site may play an important role in the hospital care and attention provided to incoming patients, improving the diagnosis accuracy and/or reducing the probability of medical errors or omissions and/or the provision of less than optimal medical services. Thus, the inclusion of these operational capacity variables is part of the innovation of our research.

The rest of this paper is structured as follows: Section 2 presents the dataset. Section 3 provides a description of the proposed methodology used in the empirical section and we present the performance of the four machine learning methodologies on our data. Section 4 provides the conclusions, a discussion on the findings and suggests possible future research based on the results of this study.

## 2. The Data

Τhe data for this study were collected from the information system of the General Hospital of Komotini “Sismanogleio” in Greece. The data records were completely anonymized, and each patient was referenced by a random code. The data were collected from 3 information data sources of the hospital information system: the patient administration system, the business intelligence system, and the laboratory information system. The data spans from 1 January 2018 to 31 December 2019. We compiled a dataset of 25 variables, that includes administrative/demographic variables, medical-clinical variables, and operational status data of the hospital. More than 20,973 hospitalizations occurred within this period. Nonetheless, 46.7% of these cases were incomplete records, missing at least one variable. We chose to eliminate these incomplete records from our dataset and work only with full ones, since the methodologies we used require datasets with no missing values (N/As) in order to work properly and efficiently. As a result, we ended up with a dataset of 11,172 cases. In 1741 of these cases (15.6%) the patient was readmitted to the hospital within the next 30 days of his last discharge.

We modelled the readmission occurrence using a binary dummy dependent variable, where 1 refers to the readmission, and 0 refers to the opposite case. In Table 1, we present the 24 explanatory variables of our dataset. More specifically, we considered 11 variables of general information for each patient (Panel A), three clinic operational status variables (Panel B), and ten explanatory variables that pertain to laboratory results from tests that were conducted on the first day of the patient’s hospitalization (Panel C).

The binary dummy variable clinic change takes the value of 1 when there is a difference between the admission and discharge clinics, and 0 to the opposite case. The past hospitalization variable indicates whether the patient was previously admitted to the same hospital within the past 30 days. Most of the variables required a series of preprocessing steps such as cleaning and filtering, using categorization techniques such as one-hot-encoding to convert integer variables to binary ones. Some of the numerical variables required a data normalization process, as each of the independent numerical variables take values on a different range.

## 3. Methodology

### 3.1. Support Vector Machines

The support vector machines [23] is a supervised machine learning methodology based on statistical learning developed for both classification and regression tasks. The support vector classification is used to separate the data into two classes by finding a hyperplane that maximizes the distance between the two classes. The hyperplane is defined by a small subset of data points, called support vectors (SV), identified through a minimization procedure (see Figure 1).

The initial dataset is split into two subsets: the training set (in-sample) and the testing set (out-of-sample). Most of the data is used in the training process, where the separating hyperplane is defined. In the testing set, the generalization ability of the model is tested on the small part of the dataset that was set aside during the training step.

#### 3.1.1. Kernel Methods

Real-life phenomena are often nonlinear and cannot be adequately described by linear classifiers (hyperplanes). To solve this drawback, the SVM paradigm is paired with kernel functions, projecting the data from the initial input space to a higher dimension space (called feature space) where the two categories may be separated linearly (Figure 2). When the kernel function is non-linear, the resulting SVM model is non-linear as well.

In this paper, we tested two kernel specifications: the linear kernel and the nonlinear radial basis function (RBF) kernel. The linear kernel detects the separating hyperplane in the original dimensional space of the data set, while the RBF displays the original data set in a higher dimensional space. The mathematical representation of each kernel is the following:(1)$$\mathrm{Linear}:\;K\;(x_{i},\;x_{j})=x_{i}^{T}x_{j}$$
(2)$$\mathrm{RBF}:\;K\;(x_{i},\;x_{j})=e^{\left(-\gamma \|x_{i}-x_{j}\|{}^{2}\right)}$$
where *γ* is the internal hyper-parameter of the RBF kernel that needs to be optimized.

#### 3.1.2. Over-Fitting

A common issue encountered during the training step is overfitting, occurring when the model “learns” to accurately describe the training data while performing worse in the test set. This is referred to in the relevant literature as the “low bias—high variance” issue. The issue is often treated using a cross-validation framework. Based on this technique, the initial train dataset is split into k equal-sized folds (parts). The training step is performed k times using a different fold each time for testing and the rest k-1 folds for training. This procedure is repeated k times with the same set of parameters until all folds have passed through the testing procedure. The performance of the model for the specific set of hyperparameters is evaluated by the average accuracy over all k test folds. An example of a three-fold cross-validation scheme is presented in Figure 3. In this study, we employ a 5-fold cross-validation procedure. Finally, the model’s generalization ability is tested using its out-of-sample accuracy on the 10% of the data that were not used in the training step.

#### 3.1.3. Weights

Our unbalanced dataset contains five times more non-readmitted cases than readmissions. We restore this imbalance by assigning inversely proportional weights to the misclassification cost.

### 3.2. Random Forests

The random forests algorithm combines the concept of decision trees with the bootstrapping and aggregating algorithm that is usually called bagging [24]. This method addresses the problem of overfitting that may occur in decision trees by combining many decision trees together into one setup called random forest.

Each tree uses a randomly selected replacement subsample of size n (equal to the size of the initial dataset). The observations that were not selected in the bootstrapping process form the out-of-bag (OOB) set. These are used to test the generalization ability of the trained model. To reduce the dependence of the models on the training set, each tree uses a randomly selected subset of the explanatory variables (features). Normally, we use the square root of the total number of features. The system aggregates the classification of each tree and retains the most popular class.

Datasets with imbalanced classes can be treated using weights, as in the SVM case, or by under-sampling the majority class or oversampling the minority class, yielding balanced datasets. The latter case is often termed the balanced random forest model.

### 3.3. Performance Metrics

In order to demonstrate the detailed forecasting performance of the machine learning classification models, we produce the confusion matrix, as illustrated in Table 2. The confusion matrix provides a detailed summary of the predictive ability of a model.

Based on the results of the confusion matrix, the following performance metrics can be calculated to evaluate the models.
(3)$$\mathrm{Recall}=\frac{TP}{TP+FN}$$
(4)$$\mathrm{Accuracy}=\frac{TP+TN}{TP+FN+FP+FN}$$
(5)$$\mathrm{Precision}=\frac{TP}{TP+FP}$$
(6)$$F1\text{-}\mathrm{Score}=2\times \frac{Precision\times Recall}{Precision+Recall}$$

The values of recall, accuracy, precision and F1-score range from 0 to 1. The higher the values, the better the model. We chose the recall (sensitivity) to be used as the key performance metric to evaluate and compare the ML models since there is a significant imbalance between the two classes of the target variable. As far as the quality of services of a hospital is concerned, we are more interested in predicting the positive data class, i.e., the cases of readmissions, than the negative cases. Therefore, the effort is focused on predicting all true positives, while minimizing false negatives.

In classification problems, accuracy is considered a significant performance metric as well. However, for unbalanced datasets, a high value of accuracy can be a misleading factor since the models tend to choose the majority class—achieving extremely high accuracy but at the same time having little success in predicting the minority class. This effect is often called the “Accuracy Paradox” [25].

Precision is the fraction of true positive cases among all cases predicted to be positive, while the F1-score is the harmonic mean of precision and sensitivity.

The area under the receiver operating characteristic curve (AUC) measures the probability that a model will rank a randomly selected positive instance higher than a randomly selected negative instance. This metric ranges from 0.5 to 1.0. When the value tends to be towards 1, it indicates a perfect ability to distinguish the different classes, while when the value tends to be towards 0.5, it means that it does not have a better performance than random guessing.

### 3.4. Empirical Results

#### 3.4.1. SVM Models

We applied a coarse-to-fine grid search scheme on the training set in order to find the optimal values of the hyperparameters that maximize the predictive ability of the SVM models. We used a 5-fold cross-validation process to avoid over-fitting. Throughout the grid search procedure, we took into account the unbalanced nature of our dataset by applying weights to the misclassification cost. The procedure continues until the parameters of the optimal model are identified.

Table 3 summarizes the results and depicts the hyperparameters of the two SVM kernels that maximize the recall.

The generalization ability of the trained model is evaluated using the testing dataset, including 1118 observations, out of which 952 refer to non-readmitted patients and 166 to readmitted ones. As performance evaluators, we use the metrics of recall (sensitivity), accuracy, precision, F1-score, and AUC. The results have been summarized in Table 4, Table 5 and Table 6.

#### 3.4.2. Random Forest Models

Similarly, to the procedure we followed concerning the SVM models, we apply a coarse-to-fine grid search evaluation scheme on the training set in order to find the optimal value of the hyperparameters that maximize the predictive capacity ability of the random forest models. In this study, we limit our search to the optimal value of the total number of decision trees.

The generalization ability of the trained models is tested using the out-of-bag (OOB) process.

Table 7 summarizes the results and depicts the optimal hyperparameters on recall (Sensitivity) for the two random forest models that we tested.

The performance of the random forest models is summarized in Table 8, Table 9 and Table 10.

The weighted random forest algorithm produced a very low sensitivity of 0.25. Thus, the forecasting ability of this model with respect to the class of interest, readmissions, is very low; it correctly predicts only 25% of all readmissions. The accuracy metric is high at 0.88, but this result is driven by the imbalanced nature of the dataset: The model forecasts the more prevalent class quite well, no readmissions, and the sparse class poorly (25%), the one of more interest, the actual readmissions. Nonetheless, the precision is high at 80%. Thus, of the cases that were forecasted as readmissions, 80% of those were correctly predicted. The rate of false alarms (1-specificity) is 1%.

The balanced random forest algorithm achieves a significantly higher sensitivity of 0.70, i.e., it correctly predicts 7 out of 10 hospitalizations whose patients will need to be readmitted. The overall accuracy is similar at 73%, while the precision is 32%. This means that, of the cases that the model forecasts as readmissions (true positives + false positives), 32% are actual readmissions (true positives). According to these results, this model manages to improve the identification of readmission cases from 25% to 70%, at the expense of a higher rate of false alarms (false positives). The false alarm rate in this case is 26% (1-specificity).

For the balanced random forest model, we estimated the impurity-based feature importance to identify the individual effect level of each feature (Table 11). The importance of a feature is computed as the (normalized) total reduction of the criterion brought by that feature. It is also known as the Gini importance.

According to the results, the two most important features are the coded diagnosis on admission and at discharge. They reveal that, as expected, some diseases (proxied by their diagnosis) have a higher probability of readmission than others. The next one, however, was quite intriguing: the clinic’s occupation rate—an operational variable—is the third most important feature in readmission forecasting. It is more important than patient age and length of stay. Thus, our innovative approach to include such variables as well uncovers their importance in forecasting readmissions. The other two operational variables, clinic’s number of doctors and clinic’s number of nurses, are somewhat important as they are ranked in the middle of the list. This is a finding that should be further investigated in a follow-up study and considered seriously by hospital management. The rest of the feature list follows without any great surprises.

#### 3.4.3. Comparative Results

Figure 4 summarizes the results of the four forecasting methodologies used. The SVM-linear and the SVM-RBF models have a sensitivity of 0.59 and 0.60, respectively. Thus, they managed to successfully predict approximately 59% and 60% of readmission cases, respectively. The existence of a large number of false positives is reflected in the precision metric, which is quite low at 0.31. Thus, the SVM models exhibit good sensitivity but an increased rate of false alarms with respect to readmissions.

The 88% accuracy of the weighted random forest model is high, but it is misleading since it predicts only 25% of the readmissions, the minority class that we are interested in.

According to these results, the best model is the balanced random forest, with a sensitivity of 0.70, which is far higher than the other three models. Also, in terms of the precision metric, the balanced random forest models slightly outperform the two SVM ones.

Concerning the area under the curve (AUC), the performance of the models is similar and consistent with previous results in the Greek and international literature (Figure 5).

## 4. Conclusions and Future Work

Predicting patients’ readmissions to hospitals can improve the quality of services provided by a health care system. It could also increase the effectiveness of initial treatment at hospitals, saving a lot of lives. The study of readmission rates may enhance discharge processes by providing better medication reconciliation and follow-up processes for discharged patients. In this study, we focused on forecasting hospital readmissions based on real-world hospital data of inpatients. The object was to build a model to predict the risk of patients’ readmission within 30 days of their index discharge.

To do this, we collected and combined a large dataset comprised of administrative/demographic data, medical-clinical data, and operational data regarding hospitalizations in the General Hospital of Komotini, Greece, between 2018 and 2019. The dataset included 11,172 cases, divided into two subsamples: in-sample and out-of-sample. Four different machine learning techniques—support vector machine with linear kernel, support vector machine with RBF kernel, weighted random forest, and balanced random forest—were trained using the in-sample dataset. For each ML model, the optimal values of the respective hyperparameters were initially found using five-fold cross-validation and out-of-bag methods to avoid overfitting. The generalization ability to unknown data was evaluated for each model using the out-of-sample dataset in terms of the sensitivity and AUC performance metrics.

The experimental results demonstrated that the balanced random forest model achieved the best performance for identifying readmission cases. Although all models managed to produce similar results in terms of the AUC metric, the balanced random forest was found to have higher sensitivity and better generalization ability than the other proposed algorithms.

Our results are promising and the proposed algorithms could be applied to a larger sample size from different source hospitals. Future work should also focus on applying other machine learning algorithms using this real-world dataset.

## Figures and Tables

**Figure 1 healthcare-10-00981-f001:**
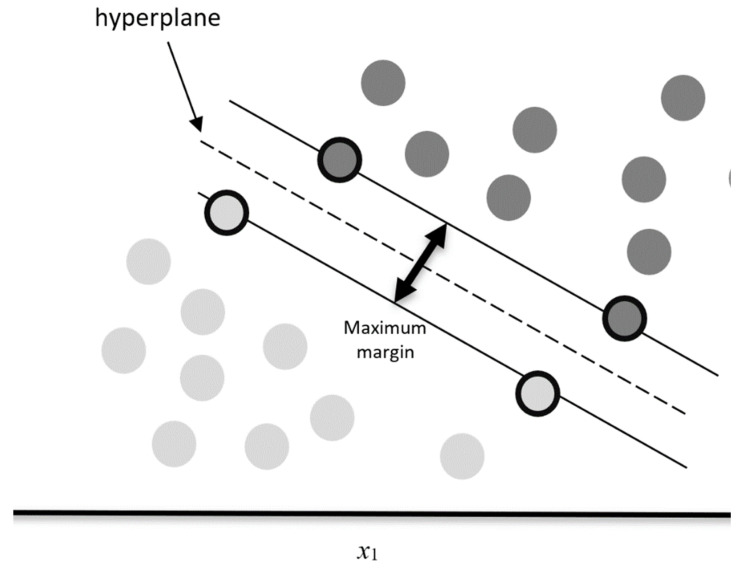
Hyperplane selection and support vectors. The pronounced black contour represents the SVs thus defining the margins with the dashed lines. The plain single line describes the separating hyperplane.

**Figure 2 healthcare-10-00981-f002:**
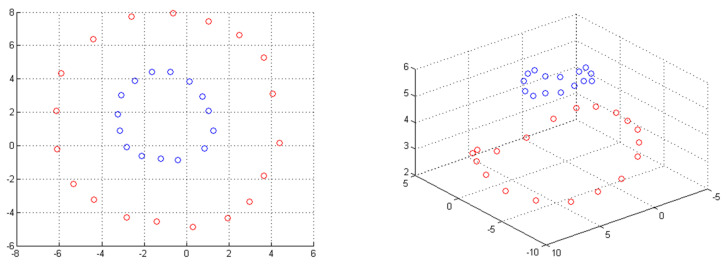
The non-separable two-class scenario in the input space(**left**) and the two-dimensional data space in a three-feature space after the projection (**right**). The two classes are represented by the different colors: blue and red.

**Figure 3 healthcare-10-00981-f003:**
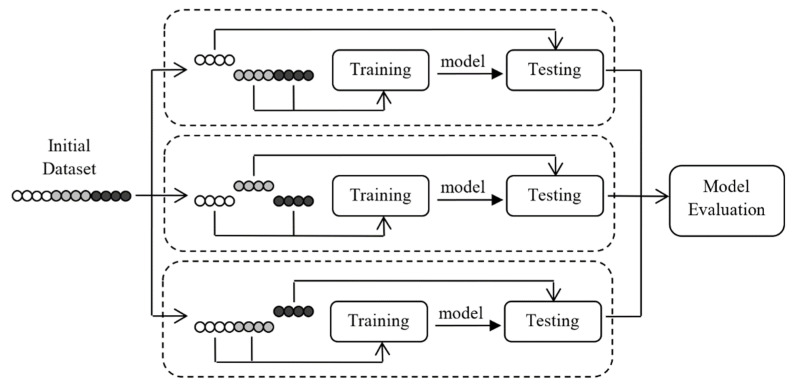
Overview of a 3-fold Cross Validation training scheme. It shows that each fold is used as a testing sample, while the remaining folds are used for training the model for each parameters’ value combination.

**Figure 4 healthcare-10-00981-f004:**
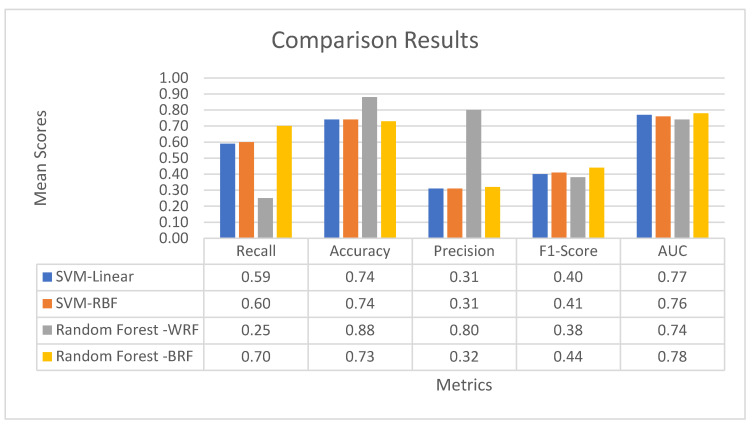
Aggregated results and comparison of proposed methodologies.

**Figure 5 healthcare-10-00981-f005:**
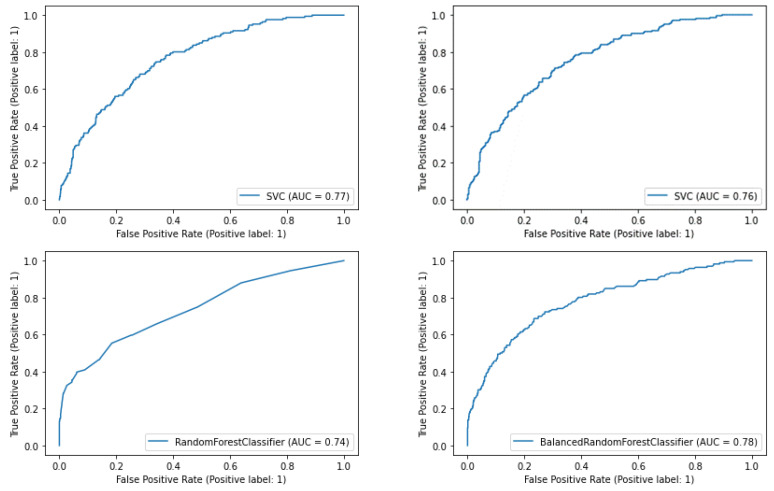
Classification performance measurement (AUC).

**Table 1 healthcare-10-00981-t001:** Input Variables of the Dataset.

No	Independent Variables	Characterization of Each Variable
**Panel A: General Information/Patient Data**		
1	Patient Age	Quantitative variable, Integer
2	Patient Gender	Qualitative variable, Categorical
3	Length of Stay	Quantitative variable, Integer
4	Patient Transfer	Qualitative variable, Binary
5	ICD-10 Diagnosis on Admission	Qualitative variable, Categorical
6	ICD-10 Diagnosis at Discharge	Qualitative variable, Categorical
7	Admission Clinic	Qualitative variable, Categorical
8	Discharge Clinic	Qualitative variable, Categorical
9	Clinic Change	Qualitative variable, Binary
10	Hospitalization Outcome	Qualitative variable, Categorical
11	Past Hospitalization	Qualitative variable, Binary
**Panel B: Operational Status of the Clinic**		
12	Clinic’s Occupancy Rate	Quantitative variable, Continuous
13	Clinic’s Number of Doctors	Quantitative variable, Integer
14	Clinic’s Number of Nurses	Quantitative variable, Integer
**Panel C: Laboratory results**		
15	Blood Sugar (Glucose)	Quantitative variable, Continuous
16	Indication (Normal Range) Blood Sugar	Qualitative variable, Categorical
17	Potassium	Quantitative variable, Continuous
18	Indication (Normal Range) Potassium	Qualitative variable, Categorical
19	Sodium	Quantitative variable, Continuous
20	Indication (Normal Range) Sodium	Qualitative variable, Categorical
21	Blood Urea Nitrogen	Quantitative variable, Continuous
22	Indication Blood Urea (Normal Range) Nitrogen	Qualitative variable, Categorical
23	Blood Creatinine	Quantitative variable, Continuous
24	Indication (Normal Range) Blood Creatinine	Qualitative variable, Categorical

**Table 2 healthcare-10-00981-t002:** Classification Results using a confusion matrix. True positives (TP)—number of samples correctly classified as readmissions. True negatives (TN)—number of samples correctly classified as non-readmissions. False positives (FP)—number of samples incorrectly classified as readmissions. False negatives (FN)— number of samples incorrectly classified as non-readmissions.

		Predicted	
		0	1
Actual	0	TN (True Negatives)	FP (False Positives)
	1	FN (False Negatives)	TP (True Positives)

**Table 3 healthcare-10-00981-t003:** Optimal parameters of SVM Models.

	Parameter C	Parameter γ
SVM Linear Kernel	0.06	---
SVM RBF Kernel	194.38	0.0001

**Table 4 healthcare-10-00981-t004:** Confusion matrix of SVM Model with linear kernel.

Confusion Matrix (SVM, Linear Kernel)			
		Predicted	
		0	1
Actual	0	TN 732	FP 220
	1	FN 68	TP 98

**Table 5 healthcare-10-00981-t005:** Confusion matrix of SVM Model with RBF Kernel.

Confusion Matrix (SVM, RBF Kernel)			
		Predicted	
		0	1
Actual	0	TN 730	FP 222
	1	FN 67	TP 99

**Table 6 healthcare-10-00981-t006:** Performance metrics of SVM Models.

SVM Linear Kernel				
Recall	Accuracy	Precision	F1-Score	AUC
0.59	0.74	0.31	0.40	0.77
SVM RBF Kernel				
Recall	Accuracy	Precision	F1-Score	AUC
0.60	0.74	0.31	0.41	0.76

Both SVM kernels produced similar results considering all performance metrics.

**Table 7 healthcare-10-00981-t007:** Optimal Parameters of random forest models.

	Total Number of Decision Trees
Weighted Random Forest	25
Balanced Random Forest	730

**Table 8 healthcare-10-00981-t008:** Confusion matrix of weighted random forest model.

Confusion Matrix (Weighted Random Forest)			
		Predicted	
		0	1
Actual	0	TN 942	FP 10
	1	FN 125	TP 41

**Table 9 healthcare-10-00981-t009:** Confusion matrix of balanced random forest model.

Confusion Matrix (Balanced Random Forest)			
		Predicted	
		0	1
Actual	0	TN 704	FP 248
	1	FN 50	TP 116

**Table 10 healthcare-10-00981-t010:** Performance metrics of random forest models.

Weighted Random Forest					
Recall	Specificity	Accuracy	Precision	F1-Score	AUC
0.25	0.98	0.88	0.80	0.38	0.74
Balanced Random Forest					
Recall	Specificity	Accuracy	Precision	F1-Score	AUC
0.70	0.74	0.73	0.32	0.44	0.78

**Table 11 healthcare-10-00981-t011:** Feature importance ranking, the significance of each feature in the classification of the random forest model in decreasing order.

Importance	Feature
0.141501	ICD-10 Diagnosis at Discharge
0.129996	ICD-10 Diagnosis on Admission
0.059492	Clinic’s Occupancy Rate
0.056195	Hospitalization Outcome
0.05464	Blood Urea Nitrogen
0.054075	Patient Age
0.052316	Potassium
0.051854	Blood Sugar (Glucose)
0.048263	Length of Stay
0.043971	Blood Creatinine
0.042001	Sodium
0.030861	Discharge Clinic
0.03023	Clinic’s Number of Doctors
0.029398	Clinic’s Number of Nurses
0.024448	Indication (Normal Range) Blood Sugar
0.020731	Admission Clinic
0.020721	Patient Gender
0.020681	Past Hospitalization
0.019997	Indication (Normal Range) Blood Creatinine
0.019523	Indication (Normal Range) Potassium
0.016082	Indication Blood Urea (Normal Range) Nitrogen
0.01222	Patient Transfer
0.01015	Indication (Normal Range) Sodium
0.00112	Clinic Change

## Data Availability

Not applicable.

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
