# Peer review of "Forecasting Hospital Readmissions with Machine Learning"

_healthcare, 2022, doi:10.3390/healthcare10060981_

Round 1

Reviewer 1 Report

This paper proposes four machine learning methods to forecast readmissions. The idea of the paper and the proposed method is in overall good and the presented results seems to be promising. However, there still some major issues that authors should revise and modify in order to accept the paper.   - There are various hospital readmission prediction methods, the authors are suggested considering some well-known and highly cited works such as 10.1016/j.dss.2022.113747,10.1016/j.knosys.2022.108326, 10.1016/j.eswa.2021.114791, 10.1016/j.neucom.2020.08.064, 10.1016/j.asoc.2020.106690, 10.1016/j.knosys.2020.106020.   - In the manuscript, authors said that 46.7% of these cases were incomplete records, missing at least one 124 variable, and eliminate these incomplete records from dataset and work 125 only with full ones. The reason of these operation should be given.   - Readmission prediction is such an intrinsically class-imbalance problem. In the manuscript, 15.6% of the patient was re-admitted to the hospital within the next 30 days. The author suggested to comparison with other class-imbalance learning methods in the experiment. I feel that the authors have failed in this part.   - Few references and most of them are old. new high-quality references should be added.    

Reviewer 2 Report

The authors evaluated the predictive performance of various machine learning models to forecast 30-day readmission after hospital discharge. They used hospital administrative data and creating the predictive models using various machine learning algorithms. Then, they tested their predictive performance metrics and compared among them. They concluded Random forest model was the best given better sensitivity and higher AUS than others.

1) Please describe the rationale to select 24 variables included in the models. Specifically, why did they select the lab results on the Day 1 of admission rather than just prior to the initial discharge?

2) What is the significance of this study? Please discuss the implication of their findings in the context of the previous literatures.

Reviewer 3 Report

This manuscript focuses on forecasting hospital readmissions with four types of machine learning. There are some defects in it.

1.In line 14, "11.172" should be replaced by "11,172".
2.In line 97, the Balanced Random Forests are used. In line 98, the problem of the imbalance is considered. But, balance and imbalance are inconsistent.
3.In lines 127 and 129, "re-admission" is used. In the other lines, "readmission" is used. Using one of the two words is better.
4.In line 136, "variable" should be replaced by "variables".
5.In lines 225-228, (1), (2), (3),and (4) should be replaced by (3), (4), (5), and (6), respectively.
6.What is the parameter "C" in Table 3?
7.In lines 267-269, the authors claim that the SVM RBF Kernel model approximates a linear kernel one because the parameter r tends to zero. However, the parameter C is much different for the two models. Why do they have similar performance? Or, is the value of C unimportant?
8.In page 2, the Random Forest and Extreme Gradient Boosting algorithms (2020), the Neural 78 Networks and the Boosted Trees (2021), and the K-Nearest Neighbors algorithm (2018) are all said to have good performance. Why are only four four types of machine learning (Support Vector Machines with Linear Kernel, Support Vector Machine with RBF (Radial Basis Function) kernel, Weighted Random Forests, and Balanced Random Forests) used here?

In conclusion, major revision is needed.

Round 2

Reviewer 2 Report

The authors revised their manuscript according to my comments, and I think their responses were appropriate and reasonable.

Author Response

We thank the reviewer for his valuable work that elevated our paper.

Reviewer 3 Report

This revised manuscript is ready to be published.

Author Response

(The authors gave the same response as above.)
